# Consideration for Flavonoid-Containing Dietary Supplements to Tackle Deficiency and Optimize Health

**DOI:** 10.3390/ijms24108663

**Published:** 2023-05-12

**Authors:** Julia Solnier, Chuck Chang, Joseph Pizzorno

**Affiliations:** 1ISURA, Clinical Research Unit, 101-3680 Bonneville Place, Burnaby, BC V3N 4T5, Canada; cchang@isura.ca; 2IMCJ, Eagan, MN 55122, USA; mail2@drpizzorno.com

**Keywords:** flavonoids, polyphenols, dietary supplements, nutrient deficiencies, drug-flavonoid interactions

## Abstract

Randomized clinical trials (RCT) and observational studies have highlighted the importance of flavonoid consumption for human health. Several studies have associated a high intake of dietary flavonoids with (a) enhanced metabolic and cardiovascular health, (b) enhanced cognitive and vascular endothelial functions, (c) an improved glycemic response in type 2 diabetes mellitus, and (d) a reduced risk of breast cancer in postmenopausal women. Since flavonoids belong to a broad and diverse family of polyphenolic plant molecules—with more than 6000 compounds interspersed in the human diet—researchers are still uncertain whether the intake of single, individual polyphenols or a large combination of them (i.e., synergistic action) can produce the greatest health benefits for humans. Furthermore, studies have reported a poor bioavailability of flavonoid compounds in humans, which presents a major challenge for determining their optimal dosage, recommended intake, and, consequently, their therapeutic value. Especially because of their scarce bioavailability from foods—along with the overall declining food quality and nutrient density in foods—the role of flavonoid supplementation may become increasingly important for human health. Although research shows that dietary supplements can be a highly useful tool to complement diets that lack sufficient amounts of important nutrients, some caution is warranted regarding possible interactions with prescription and non-prescription drugs, especially when taken concurrently. Herein, we discuss the current scientific basis for using flavonoid supplementation to improve health as well as the limitations related to high intakes of dietary flavonoids.

## 1. Introduction

Flavonoids, more specifically dietary flavonoids, are polyphenolic phytochemicals that form an important part of our daily diet (Figure 1). They occur throughout the whole plant kingdom and, thus, can be found in large quantities in plant-based foods such as fruits (e.g., apples, berries, and citrus fruits) and leafy vegetables. Flavonoids belong to a class of secondary metabolites known as low-molecular-weight phytonutrients that are present in all plant tissues [1]. Chalcones, flavones, flavonols, and isoflavones are just a few of the more than 6000 unique compounds that make up this complex category and are responsible for the brilliant colors of fruits, vegetables, herbs, and medicinal plants. In addition to their colorful functions, flavonoids also play defensive roles, such as defending plants from diverse biotic and abiotic threats, acting as UV filters, signaling molecules, phytoalexins, detoxifying agents, and antimicrobial defense compounds [1]. Scientific evidence demonstrates the importance of daily flavonoid intake for good health [2]. Yet, no Dietary Reference Intake (DRI) has been established for flavonoid components [3,4], which complicates the question of how much is needed to prevent deficiency [5]. While vitamins and minerals have been extensively researched, supplement forms of flavonoids have a shorter history due to the immense number and diversity polyphenolic phytonutrients in nature [6], which also complicates the efforts to identify these compounds, their metabolic pathways, and their physiological roles.

In this review, we discuss the relevance of flavonoid-containing dietary supplements based on (1) the overall declining food quality and related micronutrient deficiencies, and (2) the most recent clinical literature on dietary flavonoids for human health. We also discuss certain limitations regarding the high intake of dietary flavonoids.

## 2. Micronutrient Deficiencies Are Common

Micronutrient deficiencies related to vitamins and minerals currently affect more than two billion people globally. Understandably, the highest numbers are recorded in lower-income nations, but there is also a growing trend of these deficiencies in high-income countries such as the U.S., Canada, and the U.K. [7]. In North America, the obesity epidemic is ironically also dubbed as “hidden hunger” because people starve for essential nutrients while consuming foods that are high in calories [8,9]. Unfortunately, malnutrition sets off a cascade of health problems that have long-term effects on metabolic and cardiovascular health. Furthermore, it is not just the overweight or obese who suffer the effects of “hidden hunger,” because many people with normal body weights are also at risk. As can be seen in Figure 2, deficiencies of many nutrients (below estimated average requirement) are very common.

Why is this the case? At the root of this global health crisis is a lack of biodiversity in the food that we consume, or more precisely, the precipitous decimation of nutrient-rich plants which contain a wide range of valuable phytochemical compounds. Added to this is the decrease in nutrient density due to the use of synthetic fertilizers that force food to grow too fast, resulting in a dilution of nutrients. In fact, studies have shown that organically grown plant-foods contain higher amounts of health-promoting phenolic metabolites (such as flavonoids) than conventional foods [11,12,13]. Di Renzo et al. reported significant differences in the antioxidant capacities of organic foods versus conventional foods (Figure 3) and suggested that the higher total antioxidant activities of organic foods may be related to higher total contents of phytochemicals [14]. Although this hypothesis was not tested by the authors in their study, another systematic review found that organic foods contain 10–50% more phytochemicals than non-organic foods [15]. More recent studies concluded that organically grown plants, including vegetables and fruits, contain higher amounts of ascorbic acid, trace minerals, phytochemicals, phenolic compounds, and vitamins C and E, as well as fewer heavy metals, nitrates, nitrites, and pesticide residues, as well as less nitrogen [13,16,17]. Çakmakçı and Çakmakçı (2023) summarized the differences in the quality and nutritional content of organic and conventional fruits and vegetables, and concluded that various organic products (e.g., apples, apricots, asparagus, barley, carrots, strawberries, tomatoes, and onions) have higher levels of phenolic compounds [13]. According to recent studies, the higher concentrations of flavonoids and phenolic acid in organic plant-foods are related to (1) different soil management practices (mineral N-fertilizer vs organic) [18,19], (2) higher soil quality and microbial activity, as well as different fertilization practices [20,21], and (3) the response to environmental/meteorological stress conditions (e.g., temperature, humidity, and drought) [19,20,22].

Less obvious is how environmental toxins decrease a plant’s ability to absorb nutrients and increase the thirst for the nutrients that it requires. Man-made pollutants are traveling through groundwater and contaminating crops and soil. Additionally, because soil filters and stores water, promotes nutrient uptake, and protects the biodiversity of plants, few things play a greater role in growing nutrient-dense food than healthy soil [23]. Yet, across the world, complex organic nutrients in the soil are being replaced by simple chemical fertilizers, essential micronutrients are being depleted, biodiversity is being replaced with monocultures, and the levels of toxic metals such as cadmium and metalloids such as arsenic in the soil are continuing to rise. At the same time, growing global food demand, lack of sustainability, and the use of pesticides and other chemicals to increase agricultural production all contribute to the depletion of this precious resource. The result is ever decreasing nutritional values in crops and in the farm animals that feed on these crops, all because plants are not obtaining the nutrients that they need to produce the nutrients we need [24,25]. Possibly even worse, some common agricultural practices, such as tilling the soil, nitrogen fertilization [26,27], or the application of harmful synthetic chemical herbicides and pesticides [28], significantly reduce the amount of polyphenols synthesized by plants. Ultimately, the damages propagate through the entire food chain. It is no wonder that the declining quality of food is interconnected with degrading soil quality across the globe. We are only just beginning to understand that soil is a delicate and limited resource as well as a critical factor in ensuring food security [29].

As stated earlier, nutrient deficiencies that arise for whatever reason can lead to serious health problems. These include reduced immune, cognitive, and cardiovascular functions and increased susceptibility to infectious diseases [30]. One way to combat micronutrient deficiencies and enhance global nutritional resilience is to secure and expand the biodiversity of edible and nutrient-rich plant species that produce numerous phytochemical compounds such as flavonoids to positively impact human health [30]. However, due to the environmental and agricultural techniques which have already been discussed, even a balanced diet may lack certain nutrients because the foods are so depleted. To compensate, many have realized the importance of supplementing their diets, as evidenced by the continuous growth in the usage of dietary supplements globally in recent years. One reason for this might be that food quality is deteriorating as a result of increasingly highly processed calorie-rich but nutrient-poor foods. However, this is only a partial solution because preventing systemic inflammation and ensuring cardiovascular health necessitate the consumption of a high-quality and diverse diet that provides an ideal ratio of macro- and micro-nutrients, lowering the risk of developing chronic metabolic diseases [31].

A further concern is that government nutrient intake guidelines are not sufficient for those most susceptible to nutrient deficiency. These guidelines are only intended to meet the needs of the healthy majority. However, the less healthy are the group that is starving for these nutrients, and to determine sufficiency based on blood levels or intake may overlook the nutritional needs of this growing minority. Figure 4 well-illustrates the pitfalls of determining nutritional deficiency based on blood or serum levels alone. In this case, the difference between determining deficiency of vitamins B6, B12, and folate based on serum levels versus based on the toxic metabolites (homocysteine, methylmalonic acid, and 2-methylcitric acid) observed from the deficiency of these same vitamins are compared [32]. While only 6% of apparently “healthy” older adults are supposedly deficient based on vitamins detected in their blood, a surprising 63% are found to have deficiency-related toxic metabolites due to deficiency. Clearly, to avoid a buildup of deficiency-related toxic metabolites and the resulting health consequences, each person’s unique nutritional needs must be taken into account.

Can dietary supplements truly make up for a diet lacking in nutrients? Indeed, research has demonstrated that the usage of dietary supplements such as multivitamins or multiminerals is associated with fewer micronutrient deficiencies, particularly in the elderly [33,34]. Dietary supplements that include one or more isolated flavonoids can also be an important addition to diets that lack a variety of plant-based, nutrient-dense foods by lowering the risk of diseases due to micronutrient deficiencies. However, supplements cannot substitute for a high-quality, healthy, well-balanced diet containing dietary fibers and numerous other phytonutrients. Instead, supplements compensate for specific deficiencies while the broader range of nutrients and other important molecules requires more diverse sources of food.

## 3. Healthy Foods Contain Many Micronutrients, Not Just Vitamins and Minerals

Modern agriculture practices (e.g., the use of chemical fertilizers) have not only decreased the levels of micronutrients in food [27,28,35], but also dramatically decreased the levels of molecules in food that, in the past, were not considered important. Only a total of 42 nutrients are officially recognized as required for health. These include many vitamins, minerals, fatty acids, and amino acids. However, in reality, food contains as many as 50,000 elements and molecules (Figure 5) [36]. Could “experts” be correct in deciding that 99.9% of the molecules in food are unimportant? When foods are grown with synthetic chemicals, the levels of other important nutrients, such as flavonoids, plummet. Modern foods retain just enough of these other molecules to retain some of the color and flavor, but so very much else is lost. All of these other molecules contribute to the fact that organically grown foods taste so much better and are so much healthier for us [37]. Figure 6 depicts the seasonal variation of carotenoids and flavonoids when crops are grown organically and the significant loss of these nutrients when the same crops are grown conventionally with chemicals and pesticides.

## 4. Flavonoids

While it is now widely acknowledged that supplementing diets with vitamins and minerals can be beneficial, the effects of flavonoid supplementation are less well-known. In fact, flavonoids rank just behind vitamins and minerals as one of the most promising and thoroughly researched plant nutrients that promote health and decrease the risk of disease. Flavonoids have historically had a considerable role in human diets. They can be traced back to the discovery of vitamins, when early studies on scurvy discovered a connection between vitamin C and another molecule (vitamin P), as well as potential interactions between the two [39]. It was determined that the absence of vitamin C and other elements of lemon extract, which contains various flavonoid compounds, causes scurvy, or vitamin C deficiency [3]. Since then, research has linked certain flavonoids to “vitamin P (P for permeability)” activity, which is crucial for maintaining human health [3].

### 4.1. Human Research

A search on the PubMed database using the keywords “dietary flavonoids + human health” returned more than 570 results from randomized controlled trials (RCT), systematic reviews, and meta-analyses of RCT that are related to the consumption and use of dietary flavonoids and their diverse health benefits for humans. Table 1 summarizes the major clinical results from the most recent studies.

Numerous studies have demonstrated the benefits of flavonoids on cognitive health in humans [40,41,42]. Notably, a recent observational study found a link between the long-term intake of dietary flavonoids and a lower risk of cognitive decline [40]. A higher dietary intake of dietary antioxidants (e.g., anthocyanins) has been associated with a lower risk of Parkinson’s disease [43], and a cohort study associated the long-term intake of dietary flavonoids with a lower risk of Alzheimer disease and related dementia [44].

For cardiovascular health, systematic reviews and meta-analyses of randomized trials, as well as prospective cohort studies, have associated flavonoid consumption with a lower risk of cardiovascular disease mortality [41,42,45]. Flavonoid-rich diets, such as the Mediterranean or Japanese diet, have been associated with a lower risk of cardiovascular disease and diabetes, as well as longevity [46]. A Japanese study also found an inverse correlation between dietary flavonoid intake and total plasma cholesterol levels in Japanese women; the authors further associated these findings with a lower incidence of coronary heart disease in Japanese women [47]. Singh et al. reported that the Indo-Mediterranean diet (flavonoid intake 1800 mg/day)—which is rich in fruits, vegetables, and whole grains, and has a low glycemic index—can significantly reduce the incidence of coronary artery disease in high-risk patients [48]. Furthermore, study participants with type 2 diabetes mellitus showed reduced pro-inflammatory cytokine IL-6 concentrations and significantly increased plasma concentrations of naringin, hesperetin, and hesperidin when they followed a 12-week Mediterranean diet with a high intake of citrus bioflavonoids [49].

As a point of reference, the average daily consumption of flavonoids from foods in various regions of the world is as follows: 132 mg in the United States, 250–900 mg in Europe, 200–650 mg in Asia, and 1650 mg in the Middle East [46].

A recent systematic review concluded that flavonoid supplementation may help to improve the metabolic syndrome by enhancing the blood lipid profile, blood pressure, and blood glucose levels [50]. Similarly, in randomized controlled trials, dietary flavonoids have demonstrated positive effects on type 2 diabetes by reducing insulin resistance [51] and, consequently, may promote cardiovascular and metabolic health [52,53]. As a result, flavonoids have been proposed as an effective adjunct therapy to other anti-diabetic medications [52].

Additionally, several studies have highlighted wide-ranging bioactivities of flavonoid compounds which include antioxidant, anti-cancer, cardioprotective, anti-diabetic, anti-inflammatory, and anti-viral effects [1,54,55,56,57]. Apigenin has been considered as a natural remedy against chronic inflammatory conditions that are commonly associated with cancer, cardiovascular disorders, or diabetes [58,59]. Several flavonoids were found to interfere with the progression of cancer through diverse molecular mechanisms [60]. In particular, flavanones, flavones, and isoflavones can affect hormone functions by inhibiting aromatase activity and binding to estrogen receptors, and, consequently, may play an important role in hormone related cancers [61]. For instance, a recent systematic review and meta-analysis of observational studies found an association between a high intake of flavonols (e.g., kaempferol) and isoflavones (daidzein, genistein, glycitein, and formononetin) and a markedly reduced risk of ovarian cancer [62]. Along the same line, Hui et al. reported an association between a reduced risk of breast cancer and a high intake of flavonols and flavones in women [63].

The flavonol quercetin is a well-known antiplatelet compound which can inhibit platelet aggregation through various signaling pathways and, thus, may play a crucial role in the prevention of thrombosis and cardiovascular disease [64,65]. Furthermore, the anticoagulant and anti-platelet effects of quercetin can be beneficial for treating viral diseases such as COVID-19, since the infection may initiate thrombotic events through changes in platelet aggregation [66]. Thus, quercetin received attention during the COVID-19 pandemic because of its versatile anti-SARS-CoV-2 effects [67,68,69]. Clinical studies have proposed quercetin as a promising natural molecule in the preventive treatment of coronavirus infections [70], as well in the early stage of COVID-19 infections [71,72].

**Table 1 ijms-24-08663-t001:** Dietary flavonoids and their clinical benefits for human health.

Clinical Benefits	Reference
Decreased risk of coronary artery disease & cardiovascular disease mortality	Bondonno et al., 2019 [45], Micek et al., 2021 [41], Fusi et al., 2020 [42], Singh et al., 2002 [48]
Decreased risk of cognitive decline, Lower risks of Alzheimer disease & related dementias, Lower risks of Parkinson’s disease	Yeh et al., 2021 [40], Shishtar et al., 2020 [44], Talebi et al., 2022 [43]
Improvements in metabolic health (e.g., blood lipids, blood glucose and blood pressure) & insulin resistance	Gouveia et al., 2022 [50], Testa et al., 2016 [52], Al-Ishaq et al., 2019 [53], Martín and Ramos 2021 [51]
Inhibition of chronic inflammatory conditions, Reduction in pro-inflammatory cytokines e.g., IL-6	Ginwala et al., 2019 [58], DeRango-Adem and Blay 2021 [59], Al-Aubaidy et al., 2021 [49]
Reduced risk of hormone-related cancers (e.g., breast, ovarian cancer)	Liu et al., 2022 [62], Hui et al., 2013 [63]
Inhibition of platelet aggregation and thrombus formation	Tamer et al., 2022 [64], Hubbard et al., 2004 [65]
Prevention of COVID-19, Reduction in frequency & length of hospitalization, faster recovery time of COVID-19	Rondanelli et al., 2022 [70], Di Pierro et al., 2021a [71], Di Pierro et al., 2021b [72]

### 4.2. Flavonoid Bioavailability

Low oral bioavailability is yet another problem that flavonoids have in common with other bioactive plant compounds. A compound must be absorbed (i.e., “biologically accessible”) in order to exert any beneficial health effects in the body. The poor bioavailability of flavonoids makes it more difficult to determine whether a person’s dietary intake is sufficient. Furthermore, after the oral ingestion of foods or dietary supplements, flavonoids undergo extensive Phase I and Phase II metabolism in the liver. The metabolic process starts as early as when the compounds are absorbed in both the small and large intestines [73], and several conjugated forms are produced (e.g., glucurono-, sulpho-, and methyl-derivatives). These conjugated forms are the ones that pass into the blood stream. Eventually, they are deconjugated in the cells by enzymes such as β-glucuronidase and then excreted by the kidneys [42].

Numerous factors can influence bioavailability and limit clinical efficacy. For instance, differences in age, sex, genetic polymorphisms, lifestyle, dietary history, overall health state (e.g., presence of disease), and the microbiome may affect metabolic processes [73]. Hence, the bioavailability of dietary flavonoids can be highly variable between individuals—likely generating different biological responses. Furthermore, there is mounting evidence suggesting that the gut microbiota may play a key role in catabolizing flavonoids into smaller molecules (e.g., phenolic and aromatic acids), which makes them more bioavailable [73,74,75]. As a result, bioavailability is likely to be affected by the food matrix/composition, the food compound interactions, the food processing (e.g., high-temperature treatments [76]), and the plant source [42].

### 4.3. Ways to Improve Flavonoid Absorption and Clinical Benefit

Enhancing the clinical effectiveness and therapeutic potential of flavonoids requires increasing their bioavailability. The oral absorption of dietary flavonoids has been improved through several methods. Making the parent molecule more soluble and stable typically improves bioavailability by increasing the cellular absorption with amplified pharmacological effects. In a few human studies, using novel food-grade delivery systems that microencapsulate the compound in a lipomicel or liposome matrix resulted in significantly higher plasma concentrations of quercetin [77,78]. However, because flavonoids have diverse chemical structures, each flavonoid would need to undergo its own optimization for greater bioavailability.

### 4.4. Interactions

Egert and Rimbach (2011) reviewed the interaction mechanisms of flavonoids with trace elements and vitamins, as well as chemical drugs [79]. Several studies have already demonstrated that flavonoid compounds such as quercetin, as well as flavonoid-rich foods (e.g., black and green tea, coffee, red wine, and legumes), may interfere with iron absorption in the body [80]. However, iron absorption is a complex process that is affected by numerous dietary factors. This is especially true for the absorption of non-heme iron (i.e., from plant-based foods) because the process is more affected by food compounds than the absorption of heme iron from animal tissues [81,82,83]. Therefore, individuals who are already at risk of iron deficiency may need to carefully consider the high intake of dietary flavonoid-containing supplements. On the other hand, recent studies have reported that flavonoids may be beneficial as a complementary therapy to reduce iron overload due to their iron chelating and antioxidant activities [84]. Since both iron deficiency and iron overload can cause severe health issues such as hematological, metabolic, and neurodegenerative disorders, as well as carcinogenesis [85,86], flavonoids could play an essential part in regulating iron homeostasis [86].

Additionally, a few studies have discussed the interaction of flavonoid compounds with thyroid functions in terms of the synthesis and metabolism of thyroid hormones [87]. While some studies have demonstrated the antithyroid and goitrogenic effects of flavonoids, other studies found that flavonoids can stimulate the uptake of iodide [88], which is a key parameter in thyroid cancer, thus highlighting the therapeutic potential of flavonoids in radioiodine therapy [87].

Numerous in vitro drug interaction studies have investigated the inhibitory activity of flavonoids on various cytochrome P450 monooxygenase (CYP) enzymes (summarized by [79]). For example, Li et al. reported a structure-dependent inhibition of CYP3A4 by some flavonoids [89]. Šarić Mustapić et al. reported similar study results in which 7 out of 30 screened flavonoids showed significant inhibition of CYP3A4 [90]. Despite all the in vitro evidence, more definitive results from clinical research are needed to confirm these observations and determine the clinical significance of drug–flavonoid interactions.

Flavonoids can also affect drug transporters (so called “efflux pumps”) involved in the cell uptake and the extrusion of drugs and, therefore, can alter the metabolism and the absorption of drugs. In in vitro studies, flavonoids exhibited inhibitory effects on proteins which confer resistance to various conventional drugs. These include, for example, ATP-binding cassette (ABC) efflux transporters such as P-glycoprotein, breast cancer resistance protein (BCRP), and multidrug resistance-related protein 1 (MRP1) [91,92]. On the one hand, the efflux pump inhibitory activity of flavonoids can be highly beneficial for improving the clinical efficacy of otherwise poorly absorbed drugs (e.g., chemotherapeutics); on the other hand, it can alter the toxicity profile of drugs, which may be problematic for drugs with a narrow therapeutic index.

## 5. Conclusions and Future Directions

Many valuable phytochemicals, including flavonoids, may require the consumption of large amounts of edible plants in order to provide sufficient quantities to improve human health. However, due to the poor bioavailability of flavonoids, the appropriate dietary intake may vary considerably from person to person and may result in variable pharmacological responses.

In contrast to foods that provide diverse but lower levels of secondary plant metabolites, dietary supplements can complement a healthy diet by providing concentrated sources of certain nutrients and phytochemicals at levels that produce a physiological effect. As a result, they play a crucial role in both the medical and health-and-wellness industries. Given that the COVID-19 epidemic has resulted in a dramatic increase in nutritional supplement consumption around the world, there is an increased risk of opportunistic businesses advertising fraudulent, deceptive, or hazardous products. As a result, safety is an important consideration that must be assessed through meticulous and extensive quality control testing, especially since supplements are not subject to the same stringent regulations as pharmaceuticals. Furthermore, human studies (e.g., randomized controlled trials) are required in order to provide clinical and scientific evidence regarding the safe and efficacious dose ranges for these products [93]. Further, more research should be conducted on delivery mechanisms that optimize the bioavailability of diverse phytonutrients such as flavonoids to overcome the hurdles of insufficient absorption.

Additionally, more clinical studies focused on the interactions between dietary supplements (particularly those containing phytochemicals) and prescription medications are needed to ensure that concurrent use is safe. Also, the long-term use of flavonoid-containing supplements needs further investigation in future research. Only after all of this should flavonoid-based dietary supplements be considered as an option to address nutritional deficiencies and improve health.

## Figures and Tables

**Figure 1 ijms-24-08663-f001:**
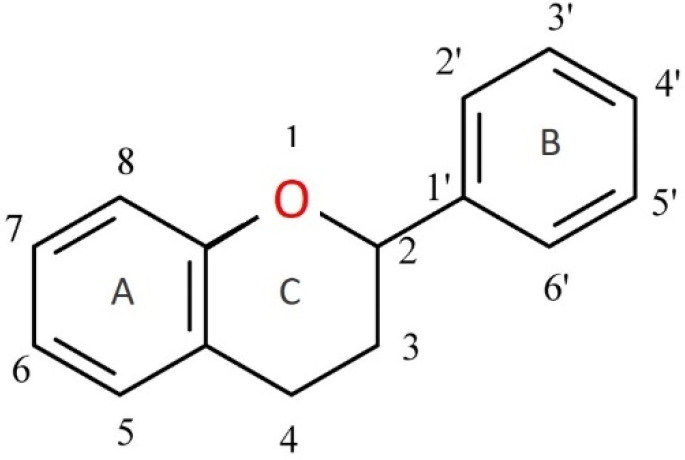
Flavan—basic skeleton structure of flavonoids.

**Figure 2 ijms-24-08663-f002:**
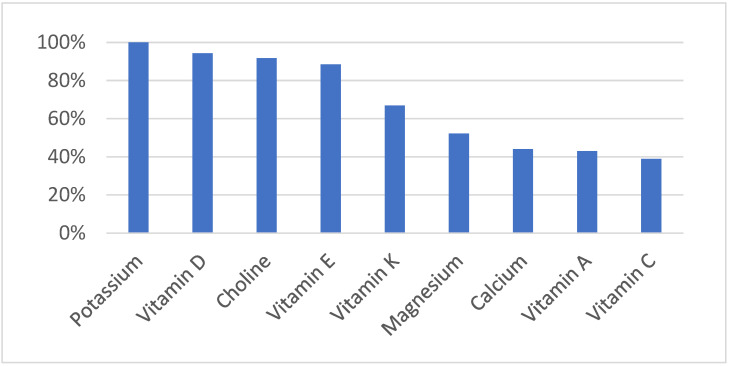
Prevalence of Nutrient Deficiency in the US [10].

**Figure 3 ijms-24-08663-f003:**
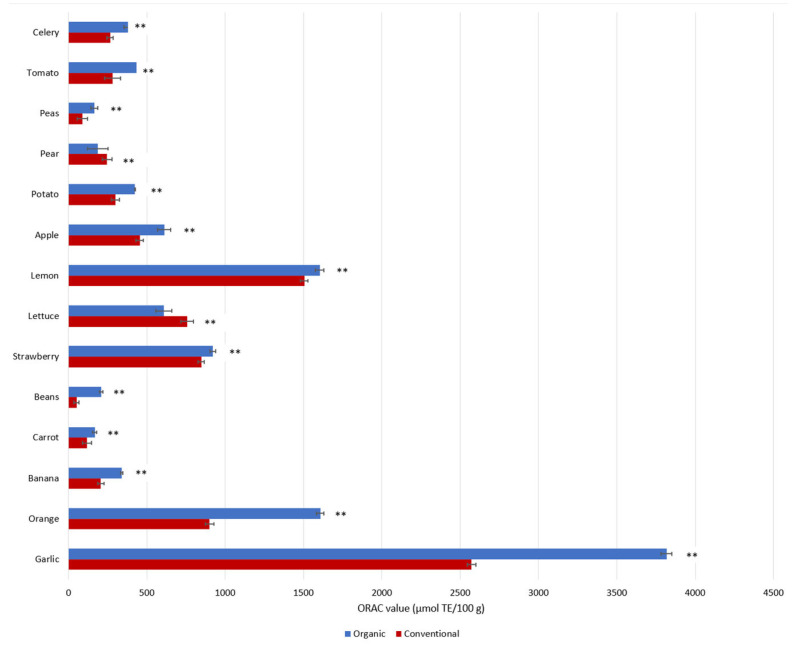
Phenolic contents as represented by antioxidant capacities (ORAC) in conventional and organic products. ORAC values are reported as medians in units of µmol Trolox Equivalent (TE) per 100 g [14]. ** *p* < 0.01, Mann–Whitney Test.

**Figure 4 ijms-24-08663-f004:**
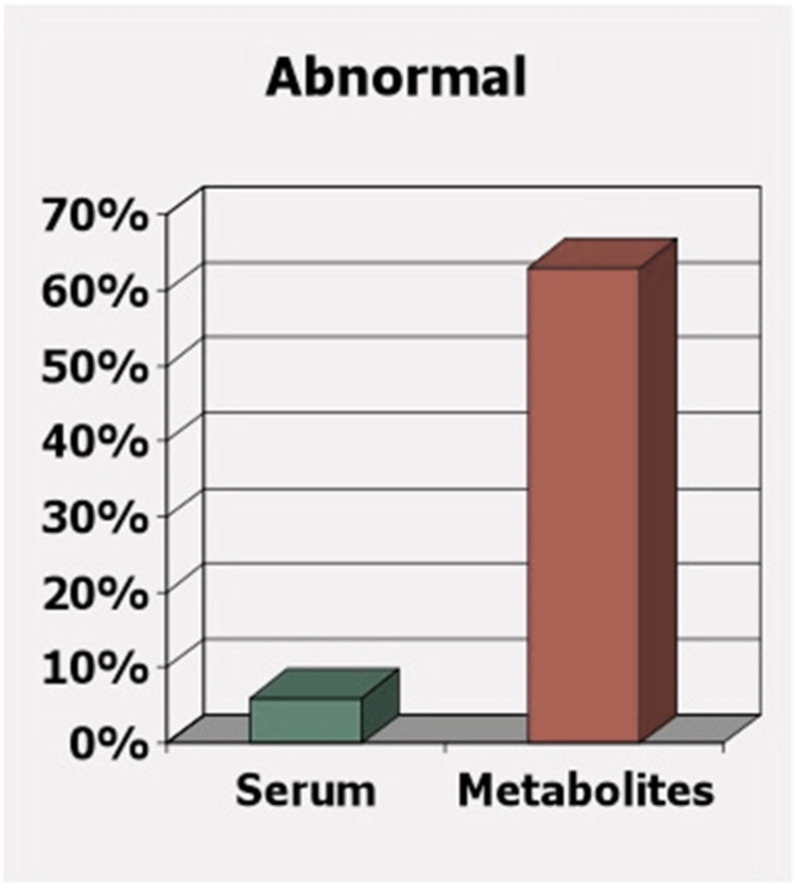
While serum levels may be “normal,” functional measures show deficiency [32].

**Figure 5 ijms-24-08663-f005:**
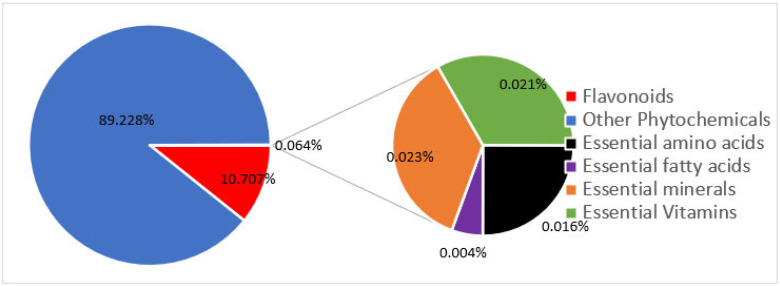
Proportion of phychemicals in foods by number. Essential vitamins, minerals, amino acids, and fatty acids make up less than 0.1% of the approximate 50,000 phytochemicals that are found in edible plants.

**Figure 6 ijms-24-08663-f006:**
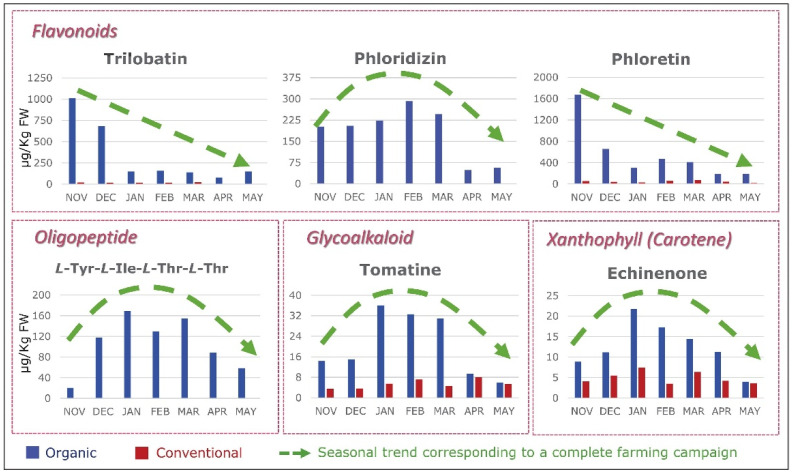
Growing Foods Chemically Greatly Decreases Their Diversity of Molecules [38].

## Data Availability

All data are publicly available.

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
