# Peer review of "Consideration for Flavonoid-Containing Dietary Supplements to Tackle Deficiency and Optimize Health"

_ijms, 2023, doi:10.3390/ijms24108663_

Round 1

Reviewer 1 Report

Dear Authors:

I reviewed your manuscript entitled "Rationale for Flavonoid-containing Dietary Supplements to Tackle Deficiency and Optimize Health". I suggest some modifications. Please, see below.

Add a graphical abstract, or a conceptual model if possible.

#6: Check authors' affiliations; make sure to include department, institute, city, state, zip code, and country.

#16: a reduced risk…  reduced risk.. Delete “a”.

#16: However, since flavonoids are a broad, … Since flavonoids are a broad,… Delete “However”.

#24: Although, research…Delete the comma.

#35: Flavonoids are widely… Revise the sentence to start in a different way. Avoid starting two sentences in a row with the exact same words. Compare with the previous sentence “Flavonoids, also known as….”..

#37: such fruitssuch as fruits…

#37: berries, citrus fruits), or leafy vegetables…. berries, and citrus fruits) or leafy vegetables.  Add “and” before “citrus”… Delete the comma before “or leafy vegetables”.

#48: Add a reference.

#51: (Di Zhou et al. 2022). … (Zhou et al. 2022).  Also, correct it in the Ref. list #410.

#53-55: Add a ref.

#63-64: Canada and the U.K. … Canada, and the U.K.  Add a comma.

#66: (Fanzo et al. 2022; Eggersdorfer et al. 2018)… (Eggersdorfer et al. 2018; Fanzo et al. 2022)

#68: And it is not… Furthermore, it is not…

#82: (Di Renzo et 82 al. 2007)…Replace with a more recent ref. if possible.

#85: Although, the authors… Delete the comma.

#88-89: Add units. Also, the fig does not show the significance level. Add the significant level and error bars.

#105: practices such as tilling the soil or application of glyphosate… practices, such as tilling the soil or applying glyphosate…

#113: and an increased susceptibility… and increased susceptibility…

#116-117: so as to have a positive impact on human health… to positively impact the human health..

#128: majority of healthy people….Unbold.

#132: of Vitamins B6, B12,… of vitamins B6, B12,…

#151: Instead, supplements are used to compensate for… Instead, supplements compensate for…

#156: this technology… What “technology” are you referring to? Consider revising this word.

#155-156: Add references.

#163: color and flavour… color and flavor…

#179: Vitamin Cvitamin C…

#180: (BRUCKNER and SZENT-GYÖRGYI… Why in all uppercase letters?

#181-182: which contains a variety of different flavonoid compounds, is what causes… which contains various flavonoid compounds, causes…

#193: (Yeh et al. 2021; Micek et al. 2021; Fusi et al. 2020)…. (Fusi et al. 2020; Micek et al. 2021; Yeh et al. 2021).

#202: Several flavonoid-rich diets such as the Mediterranean, or Japanese diet have… Several flavonoid-rich diets, such as the Mediterranean or Japanese diet, have…

#213-214: hesperitin and hesperidin… hesperetin, and hesperidin…Correct the name and add a comma.

#215-217: 132 mg per day in the United States, 250–900 mg per day in Europe, 200–650 mg per day in Asia, and 1650 mg per day in the Middle East…132 mg in the United States, 250–900 mg in Europe, 200–650 mg in Asia, and 1650 mg in the Middle East…

#221: trials dietary… trials, dietary… Add a comma.

Please, check the entire manuscript for typos like these ones, unnecessary commas and missed ones.

#230-231: conditions, which are commonly associated with cancer,… conditions, commonly associated with cancer,…

#281: through the several methods…. through several methods.

#289-290: Egert et al. reviewed the interaction mechanisms of flavonoids with trace elements, vitamins, as well as chemical drugs (Egert and Rimbach 2011). .. Egert and Rimbach (2011) reviewed the interaction mechanisms of flavonoids with trace elements, vitamins, as well as chemical drugs.

#296: more affected by food compounds compared to the absorption of heme iron… more affected by food compounds than the absorption of heme iron…

#305: (Lesjak and K S Srai 2019)… Use only author’s last name here, no initials.

#309: other studies found that flavonoids can stimulate… Cite some of those studies, not only the review article.

#335: at levels which produce… at levels that produce…

#359: Cancers 11 (1). … Cancers 11, 28.

#358-570: Rewrite following the journal style for reference list. Also, follow journal style for citing ref. in the text.

Please, see my comments which include some suggestions for improvement.

Add a graphical abstract, or a conceptual model if possible.

#6: Check authors' affiliations; make sure to include department, institute, city, state, zip code, and country.

#16: a reduced risk…  reduced risk.. Delete “a”.

#16: However, since flavonoids are a broad, … Since flavonoids are a broad,… Delete “However”.

#24: Although, research…Delete the comma.

#35: Flavonoids are widely… Revise the sentence to start in a different way. Avoid starting two sentences in a row with the exact same words. Compare with the previous sentence “Flavonoids, also known as….”..

#37: such fruitssuch as fruits…

#37: berries, citrus fruits), or leafy vegetables…. berries, and citrus fruits) or leafy vegetables.  Add “and” before “citrus”… Delete the comma before “or leafy vegetables”.

#48: Add a reference.

#51: (Di Zhou et al. 2022). … (Zhou et al. 2022).  Also, correct it in the Ref. list #410.

#53-55: Add a ref.

#63-64: Canada and the U.K. … Canada, and the U.K.  Add a comma.

#66: (Fanzo et al. 2022; Eggersdorfer et al. 2018)… (Eggersdorfer et al. 2018; Fanzo et al. 2022)

#68: And it is not… Furthermore, it is not…

#82: (Di Renzo et 82 al. 2007)…Replace with a more recent ref. if possible.

#85: Although, the authors… Delete the comma.

#88-89: Add units. Also, the fig does not show the significance level. Add the significant level and error bars.

#105: practices such as tilling the soil or application of glyphosate… practices, such as tilling the soil or applying glyphosate…

#113: and an increased susceptibility… and increased susceptibility…

#116-117: so as to have a positive impact on human health… to positively impact the human health..

#128: majority of healthy people….Unbold.

#132: of Vitamins B6, B12,… of vitamins B6, B12,…

#151: Instead, supplements are used to compensate for… Instead, supplements compensate for…

#156: this technology… What “technology” are you referring to? Consider revising this word.

#155-156: Add references.

#163: color and flavour… color and flavor…

#179: Vitamin Cvitamin C…

#180: (BRUCKNER and SZENT-GYÖRGYI… Why in all uppercase letters?

#181-182: which contains a variety of different flavonoid compounds, is what causes… which contains various flavonoid compounds, causes…

#193: (Yeh et al. 2021; Micek et al. 2021; Fusi et al. 2020)…. (Fusi et al. 2020; Micek et al. 2021; Yeh et al. 2021).

#202: Several flavonoid-rich diets such as the Mediterranean, or Japanese diet have… Several flavonoid-rich diets, such as the Mediterranean or Japanese diet, have…

#213-214: hesperitin and hesperidin… hesperetin, and hesperidin…Correct the name and add a comma.

#215-217: 132 mg per day in the United States, 250–900 mg per day in Europe, 200–650 mg per day in Asia, and 1650 mg per day in the Middle East…132 mg in the United States, 250–900 mg in Europe, 200–650 mg in Asia, and 1650 mg in the Middle East…

#221: trials dietary… trials, dietary… Add a comma.

Please, check the entire manuscript for typos like these ones, unnecessary commas and missed ones.

#230-231: conditions, which are commonly associated with cancer,… conditions, commonly associated with cancer,…

#281: through the several methods…. through several methods.

#289-290: Egert et al. reviewed the interaction mechanisms of flavonoids with trace elements, vitamins, as well as chemical drugs (Egert and Rimbach 2011). .. Egert and Rimbach (2011) reviewed the interaction mechanisms of flavonoids with trace elements, vitamins, as well as chemical drugs.

#296: more affected by food compounds compared to the absorption of heme iron… more affected by food compounds than the absorption of heme iron…

#305: (Lesjak and K S Srai 2019)… Use only author’s last name here, no initials.

#309: other studies found that flavonoids can stimulate… Cite some of those studies, not only the review article.

#335: at levels which produce… at levels that produce…

#359: Cancers 11 (1). … Cancers 11, 28.

#358-570: Rewrite following the journal style for reference list. Also, follow journal style for citing ref. in the text.

Author Response

We would like to thank the reviewers and the editorial team for their constructive feedback and taking the time to review this manuscript. Please find below – our responses to address these comments.

Reviewer 1/Report1:

Dear Authors:

I reviewed your manuscript entitled "Rationale for Flavonoid-containing Dietary Supplements to Tackle Deficiency and Optimize Health". I suggest some modifications. Please, see below.

Add a graphical abstract, or a conceptual model if possible.

A graphical abstract has been added. Please see line 28.

#6: Check authors' affiliations; make sure to include department, institute, city, state, zip code, and country.

This has been revised. Please see line 5 and 6.

#16: a reduced risk…  reduced risk.. Delete “a”.

Revised – please see line 14.

#16: However, since flavonoids are a broad, … Since flavonoids are a broad,… Delete “However”.

Revised – please see line 14.

#24: Although, research…Delete the comma.

Please note that the entire manuscript has been checked for typos like these ones, unnecessary commas and missed ones. Please see line 23.

#35: Flavonoids are widely… Revise the sentence to start in a different way. Avoid starting two sentences in a row with the exact same words. Compare with the previous sentence “Flavonoids, also known as….”..

Revised – please see line 35.

#37: such fruits … such as fruits…

Revised – please see line 37.

#37: berries, citrus fruits), or leafy vegetables…. berries, and citrus fruits) or leafy vegetables.  Add “and” before “citrus”… Delete the comma before “or leafy vegetables”.

Revised – please see line 37.

#48: Add a reference.

Reference has been added. Please see line 46.

#51: (Di Zhou et al. 2022). … (Zhou et al. 2022).  Also, correct it in the Ref. list #410.

References have been updated and formatted according to the journal’s guidelines.

#53-55: Add a ref.

Reference has been added. Please see line 50.

#63-64: Canada and the U.K. … Canada, and the U.K.  Add a comma.

Revised – please see line 63.

#66: (Fanzo et al. 2022; Eggersdorfer et al. 2018)… (Eggersdorfer et al. 2018; Fanzo et al. 2022)

References have been updated and formatted according to the journal’s guidelines.

#68: And it is not… Furthermore, it is not…

Revised – please see line 67.

#82: (Di Renzo et 82 al. 2007)…Replace with a more recent ref. if possible.

More recent studies have been added. Please see line 84-96.

#85: Although, the authors… Delete the comma.

Revised – please see line 82.

#88-89: Add units. Also, the fig does not show the significance level. Add the significant level and error bars.

Figure 3 has been revised. Please see line 100: “Phenolic contents as represented by antioxidant capacities (ORAC) in conventional and organic products. ORAC values are reported as medians in units of µmol Trolox Equivalent (TE) per 100 grams. ** P < 0.01, Mann-Whitney Test.”

#105: practices such as tilling the soil or application of glyphosate… practices, such as tilling the soil or applying glyphosate…

Revised – please see line 116.

#113: and an increased susceptibility… and increased susceptibility…

Revised – please see line 125.

#116-117: so as to have a positive impact on human health… to positively impact the human health..

Revised – please see line 128.

#128: majority of healthy people….Unbold.

Revised – please see line 141.

#132: of Vitamins B6, B12,… of vitamins B6, B12,…

Revised – please see line 145.

#151: Instead, supplements are used to compensate for… Instead, supplements compensate for…

Revised – please see line 162.

#156: this technology… What “technology” are you referring to? Consider revising this word.

Revised – please see line 167 and 168.

#155-156: Add references.

References have been added. Please see line 168.

#163: color and flavour… color and flavor…

Revised – please see line 176.

#179: Vitamin C … vitamin C…

Revised – please see line 192.

#180: (BRUCKNER and SZENT-GYÖRGYI… Why in all uppercase letters?

#181-182: which contains a variety of different flavonoid compounds, is what causes… which contains various flavonoid compounds, causes…

Revised – please see line 194.

#193: (Yeh et al. 2021; Micek et al. 2021; Fusi et al. 2020)…. (Fusi et al. 2020; Micek et al. 2021; Yeh et al. 2021).

References have been updated and formatted according to the journal’s guidelines.

#202: Several flavonoid-rich diets such as the Mediterranean, or Japanese diet have… Several flavonoid-rich diets, such as the Mediterranean or Japanese diet, have…

Revised – please see line 213.

#213-214: hesperitin and hesperidin… hesperetin, and hesperidin…Correct the name and add a comma.

Revised – please see line 223.

#215-217: 132 mg per day in the United States, 250–900 mg per day in Europe, 200–650 mg per day in Asia, and 1650 mg per day in the Middle East…132 mg in the United States, 250–900 mg in Europe, 200–650 mg in Asia, and 1650 mg in the Middle East…

Revised – please see line 227 and 228.

#221: trials dietary… trials, dietary… Add a comma.

Revised – please see line 232.

Please, check the entire manuscript for typos like these ones, unnecessary commas and missed ones.

The entire manuscript has been checked for typos like these ones, unnecessary commas and missed ones.

#230-231: conditions, which are commonly associated with cancer,… conditions, commonly associated with cancer,…

Revised – please see line 240.

#281: through the several methods…. through several methods.

Revised – please see line 307.

#289-290: Egert et al. reviewed the interaction mechanisms of flavonoids with trace elements, vitamins, as well as chemical drugs (Egert and Rimbach 2011). .. Egert and Rimbach (2011) reviewed the interaction mechanisms of flavonoids with trace elements, vitamins, as well as chemical drugs.

Revised – please see line 316.

#296: more affected by food compounds compared to the absorption of heme iron… more affected by food compounds than the absorption of heme iron…

Revised – please see line 323.

#305: (Lesjak and K S Srai 2019)… Use only author’s last name here, no initials.

References have been updated and formatted according to the journal’s guidelines.

#309: other studies found that flavonoids can stimulate… Cite some of those studies, not only the review article.

Reference has been added. Please see line 334.

#335: at levels which produce… at levels that produce…

Revised – please see line 362.

#359: Cancers 11 (1). … Cancers 11, 28.

#358-570: Rewrite following the journal style for reference list. Also, follow journal style for citing ref. in the text.

The manuscript has been formatted according to the journal’s guidelines.

Reviewer 2 Report

This paper is devoted to discuss the current scientific basis for using flavonoid supplementation to improve health as well as the limitations related to high intakes of dietary flavonoids by performing review. I think the topic is important and contributive to the promotion of health for neuroscience professionals. However, before this paper is published, the following comments should be taken into account when revising the paper.

Major concerns:

1.      The authors should modify tile to match type in this study.

2.      Pros and cons of proposed review approach should be specified.

3.      The authors should register to PERSEO get an approval code.

4.      Does any basic mechanism or etiology to explain the rationale of this review study? Please provide illustrations.

5.      Why no comparisons with traditional nutrition supplement? Please give more illustrations.

6.      The contribution of this paper should be highlighted.

Author Response

We would like to thank the reviewers and the editorial team for their constructive feedback and taking the time to review this manuscript. Please find below – our responses to address these comments.

Reviewer 2/Report 2:

Major concerns:

  1. The authors should modify title to match type in this study.

Please see the revised title of the literature review: “Consideration for Flavonoid-containing Dietary Supplements to Tackle Deficiency and Optimize Health

  1. Pros and cons of proposed review approach should be specified.

In the literature review “Consideration for Flavonoid-containing Dietary Supplements to Tackle Deficiency and Optimize Health” we discuss the relevance of flavonoid-containing dietary supplements based on (1) the overall declining food quality and related micronutrient deficiencies, and (2) the most recent clinical literature on dietary flavonoids for human health. We also discuss certain limitations regarding the high intake of dietary flavonoids, which may contradict our hypothesis, but reduces the risk of bias. Furthermore, in this review we identify gaps in research and provide directions for future studies.

Cons of this review approach are: (1) since a large amount of literature exists, we only focused on the major clinical results from the most recent studies. Therefore, we searched the PubMed database using the keywords "dietary flavonoids + human health" filtered by free full text availability and randomized controlled trials (RCT), Systematic Reviews, and Meta Analysis of RCT; (2) hence this literature review is limited to published studies (with free full text availability) only, other relevant studies may be missed.

  1. The authors should register to PERSEO get an approval code.

Thank you for the suggestion! We could not register with PROSPERO because of October 1, 2019, PROSPERO no longer accepts registration of reviews where data extraction has already started. Furthermore, PROSPERO does not apply to this literature review since PROSPERO is a prospective register for systematic reviews.

  1. Does any basic mechanism or etiology to explain the rationale of this review study? Please provide illustrations.

      The topic or research question of this review was to discuss the relevance of flavonoid-containing dietary supplements based on (1) the overall declining food quality and related micronutrient deficiencies, and (2) the current clinical literature on dietary flavonoids for human health. Therefore, we summarized the major clinical results from the most recent studies by searching the PubMed database using the keywords "dietary flavonoids + human health" filtered by free full text availability and randomized controlled trials (RCT), Systematic Reviews, and Meta Analysis of RCT.

      Furthermore, we added a graphical abstract to illustrate key ideas from this review study.

  1. Why no comparisons with traditional nutrition supplement? Please give more illustrations.

Traditional nutritional supplements are typically based on providing higher doses of vitamins and minerals than that attainable through dietary sources, and much research have been done on how vitamins and minerals influence health. However, flavonoids have little therapeutic overlap with vitamins and minerals. So, it is difficult to compare the two. It is possible that future research can provide more insights where they can be compared. Figure 6 has been created to compare the number of compounds covered by traditional supplements providing essential nutrients to the number of flavonoids and other phytochemicals.

  1. The contribution of this paper should be highlighted.

In this review we summarize and highlight the importance of flavonoid compounds for human health based on the latest results of clinical trials as well as prospective cohort studies.

Furthermore, we identify and highlight knowledge gaps in the literature, and provide directions for future studies. The goal of this review was also to provide the general public and healthcare practitioners with evidence-based guidance and helping them to have a greater understanding and confidence in using flavonoid-based dietary supplements to address nutritional deficiencies and improve their health.

Please see the revised “Conclusions and Future Perspectives”. Line 355 – 379.

Round 2

Reviewer 2 Report

Thanks for great efforts on revision.